# Casting Process Improvement by the Application of Artificial Intelligence

Nedeljko Dučić [1], Srećko Manasijević [2], Aleksandar Jovičić [1,*], Žarko Ćojbašić [3] and Radomir Radiša [2]

1   Faculty of Technical Sciences Cacak, University of Kragujevac, Svetog Save 65, 32102 Cacak, Serbia; nedeljko.ducic@ftn.kg.ac.rs
2   Research and Development Institute Lola Ltd., 70A Kneza Višeslava, 11030 Belgrade, Serbia; srecko.manasijevic@li.rs (S.M.); radomir.radisa@li.rs (R.R.)
3   Faculty of Mechanical Engineering, University of Nis, Aleksandra Medvedeva 14, 18000 Nis, Serbia; zcojba@ni.ac.rs
*   Correspondence: aleksandar.jovicic@ftn.kg.ac.rs

**Abstract:** On the way to building smart factories as the vision of Industry 4.0, the casting process stands out as a specific manufacturing process due to its diversity and complexity. One of the segments of smart foundry design is the application of artificial intelligence in the improvement of the casting process. This paper presents an overview of the conducted research studies, which deal with the application of artificial intelligence in the improvement of the casting process. In the review, 37 studies were analyzed over the last 15 years, with a clear indication of the type of casting process, the field of application of artificial intelligence techniques, and the benefits that artificial intelligence brought. The goals of this paper are to bring to attention the great possibilities of the application of artificial intelligence for the improvement of manufacturing processes in foundries, and to encourage new ideas among researchers and engineers.

**Keywords:** casting process; artificial intelligence; improvement; intelligent casting

## 1. Introduction

The production of metal parts by a casting process is a very old manufacturing process, the basic principles of which have remained unchanged up until now. Thus, scientific research on the mentioned manufacturing process has been constantly expanding. As a result, numerous casting processes—as well as materials that are successfully processed by the casting process—have been developed. Metal casting products are present in all spheres of human life, from small bolts to components with various dimensions and complexity for military systems and aircraft. This is why the casting industry has become diversified, and therefore often specialized for the manufacturing of particular types of casting parts.

Nowadays, science and technology's development has lead us to what is called Industry 4.0, which actually represents a new wave of automation development in manufacturing, based on technologies such as automation, data exchanges, artificial intelligence (AI), the cloud, cyber-physical systems, robots, big data analytics, and the Internet of Things (IoT), etc. What does Industry 4.0 specifically mean for the casting process? If we consider Industry 4.0 as one set and the casting process as the other set, the cross-section of these two sets is the smart foundry. Smart foundries imply a centralized system of customer order management, supply chains, and manufacturing activities. Part of such a centralized system is remote monitoring and the control of machines and robotic systems in the foundry, based on cloud principles. One part of such a system is expert systems—based on artificial intelligence—that make decisions, optimize the design of casting systems, and define recipes (e.g., the molten metal, the sand mixture for the mold, etc.). The road to the realization of the smart foundry concept involves debt, and requires a workforce that values its importance and can accept the monitoring of the technological progress.

One of the segments of the smart foundry concept is the application of artificial intelligence techniques in the research and development of manufacturing processes in foundries. In this paper are presented, in a clear form, research studies that are related to the application of artificial intelligence techniques to the improvement of the casting process, from the last 15 years. The presented studies, 37 of them, refer to four casting technologies—sand casting, pressure die casting, continuous casting, and investment casting—as well as five artificial intelligence techniques: artificial neural networks (ANN), genetic algorithms (GA), particle swarm optimization (PSO), fuzzy logic (FL) and the adaptive network-based fuzzy inference system (ANFIS). The first goal of this paper is to bring to attention to the great possibilities of the use of artificial intelligence in the improvement of the manufacturing processes in foundries, through an accurate analysis of the presented studies. The second goal of this paper is to encourage new ideas among researchers and engineers, which will contribute to the development and implementation of the Industry 4.0 concept in foundries, i.e., the smart foundry concept.

## 2. Methods of Artificial Intelligence

Artificial intelligence and its application in manufacturing are no longer a vision of the future. Its application is widespread, and it is used for the modeling and optimization of processes, to find new solutions, for predictive maintenance, and for digital monitoring and control, etc. Some basic information on the artificial intelligence techniques that were used in the analyzed studies is given below.

### 2.1. Artificial Neural Networks—ANN

The artificial neural network (ANN) is a flexible tool, capable of modeling non-linear processes by establishing relationships between input and output process parameters—based on the appropriate amount of data—which describe the process. The work of the artificial neural network (ANN) is based on the imitation of the human brain's function, characterized by its high execution speed and high level of ability in solving complex and non-linear problems. An artificial neural network (ANN) is a mathematical representation of the human brain's function. The two most important characteristics of the artificial neural network (ANN) are the network architecture, i.e., its structure, which includes the number of hidden layers and the neurons in them, and the training algorithm. The fundamental element of the artificial neural network (ANN) is a neuron, and neurons are interconnected through synapses, as characterized by synaptic weight. The principles of their functioning, and the types and learning of the algorithms of neural networks have been a research challenge since the very beginning of the development of neural network in the mid-twentieth century, in fact from the first studies conducted by McCulloch, Pitts, Hebb, Widrow, Hoff and von Neumann, and later by Rumelhart and his research team. Nowadays, these research studies are available in various literary forms [1,2].

### 2.2. Genetic Algorithms—GA

Genetic algorithms (GA) are an efficient optimization technique that belongs to the family of evolutionary algorithms. The characteristics of this algorithm are based on the imitation of the biological evolution process. Namely, the initial population is randomly formed, and it represents a set of potential solutions, the validities of which are assessed on basis of the fitness function. The solutions with the best characteristics are transferred to the new population, and genetic operators such as selection, crossover and mutation are applied to them. In the next iterations, a new assessment and a new genetic operator's application are carried out, until the corresponding criteria of the optimization process are satisfied. In the 1960s, Holland promoted the GA as an optimization tool, and then worked with his associates on its further development, which was presented in numerous available literary sources [3–7]. Genetic algorithms, nowadays, have found applications in many engineering applications due to their ability to successfully resolve numerous optimization problems.

### 2.3. Particle Swarm Optimization—PSO

Particle swarm optimization (PSO) is a stochastic optimization technique which was developed by Eberhart and Kennedy in 1995, and was inspired by the social behavior of birds flocking and fishes schooling [8,9]. Namely, their optimization algorithm basically contains a simulation of birds flocking around food sources. The position of the food source is the solution of the optimization problem, and the locations of the individuals searching for food are potential solutions for every iteration of the algorithm. In fact, the algorithm begins its work by creating initial particles to which it assigns initial velocities. In every new iteration, the algorithm evaluates the location of each particle based on the objective function and defines the best location. Thereafter, it follows a new selection of particle velocities based on the current velocity, the individual best locations, and the best locations of their neighbors. In this way, it repetitively updates the particle locations. The algorithm ends its work when it reaches the stopping criteria [10,11].

### 2.4. Fuzzy Logic—FL

The idea of fuzzy logic (FL) was promoted by Professor Zadeh of the University of California at Berkeley in 1965 [12]. Professor Mamdani used a fuzzy logic (FL) concept to control an automatic steam engine [13]. Since the 1980s, there has been an increase in fuzzy logic's application in industrial manufacturing, automatic control, automobile production, banks, and hospitals etc. The application of fuzzy logic is based on steps such as fuzzification, the fuzzy inference process, and defuzzification. Fuzzification is a process that involves the conversion of classical data into a form of fuzzy data or membership functions (MFs). The fuzzy inference process generates fuzzy output based on a combination membership functions and fuzzy rules. Defuzzification is a process that transforms a fuzzy conclusion or output, which is a linguistic variable, into a crisp variable [14].

### 2.5. Adaptive Network-Based Fuzzy Inference System—ANFIS

The adaptive network-based fuzzy inference system (ANFIS) is a specific neuro-fuzzy technique which produced the fusion between the neural network and the fuzzy inference system [15–17]. Fuzzy logic, as part of the ANFIS methodology, considers the inaccuracy and uncertainty of the system being modelled, and the neural network provides the possibility of the adaptation and adjustment of the fuzzy performance. Thus, the neural network is an adaptation mechanism that can perform the modification of control rules, the modification of the primary fuzzy sets, and the selection of the defuzzification method, etc. The application of ANFIS in manufacturing technologies is reflected in the modeling of different non-linear processes, and in the use of this methodology in the design of control systems.

## 3. Literature Review

### 3.1. Improvement of the Sand Casting Process

The sand casting process is a very frequently used casting process that provides the possibility of producing a variety of metal components with complex geometries. The produced parts can vary greatly in size and weight, ranging from a couple of ounces to several tons. The mold's properties have a significant influence on the quality of the sand casting process, i.e., the casting part quality. These properties are: dimensional and thermal stability at elevated temperatures, permeability, hardness, consistent cleanliness, composition, and others. All of the above-mentioned mold properties depend on the process parameters, such as the grain size and shape, the percentage of clay, and the percentage of water, etc. The mentioned dependence between the process parameters and mold characteristics is a research challenge that has existed for many years. In 1960s, Frost and Hiller defined the pressure and hardness distributions in sand molds [18]. Marek developed significant mathematical equations that established the correlation between the important parameters and quality of sand molds [19]. Wenninger provided a strong theoretical approach by explaining sand–clay–water relationships [20]. The application of artificial intelligence is

noticeable in recent research studies. Its application is reflected in the modeling of complex dependencies between input parameters and mold characteristics, as well as in the optimization of input parameters to secure a mold with the pre-defined characteristics.

Karunakar and Datta researched the definition of the optimal composition of a green sand mixture in order to obtain the desired characteristics of sand mold systems. In their study, they used neural networks and genetic algorithms to achieve the defined goal. In the first part of the study, they trained neural networks, the inputs of which were the mold characteristics (green compression strength, green shear strength, permeability, dry compression strength and dry shear strength), and the outputs of which were the process parameters (the grain fineness number, clay percentage, moisture percentage, mulled time, and hardness). The task of the developed neural network was to project the process parameters based on the desired characteristics of the mold. In the second part of the study, the fitness function for the work of the genetic algorithm was formed. The task was the same as in the first part of the study: the prediction of the process parameters with which the desired mold characteristics were obtained. When it comes to this task, genetic algorithms yielded better results [21]. In order to help detect the most common casting defects—such as cracks, misruns, scabs, blowholes, and airlocks—the same research team, Karunakar and Datta, designed an intelligent system based on back-propagation neural networks. The research was based on the utilization of an immense database, and the main objective was to upgrade the prevention features of the neural network prediction so that potential faults which were the result of the sand mold system's characteristics could be detected more easily [22]. The neural network input parameters were the composition of the charge, melting conditions, moisture percentage, green compression strength, permeability, and green shear strength.

For the optimization of a green sand mold system, Surekha et al. used particle swarm optimization (PSO) and genetic algorithms (GA). They utilized the specific optimization approach to achieve the optimal solution for a green sand mold system, by which they calculated a correlation between the process parameters such as the grain fineness number, percentage of clay, percentage of water, and number of strokes, and responses such as the green compression strength, permeability, hardness, and bulk density. Their objective was to reach a middle-ground solution that included all of the four satisfactory features: green compression strength, permeability, hardness, and bulk density [23]. Parappagoudar et al. established a correlation between the characteristics of cement-bonded sand molds and process parameters by using feed-forward neural networks trained with the help of a back-propagation algorithm and a genetic algorithm, separately. The mold characteristics, which were the subject of the prediction, were the compression strength and hardness, and the process parameters were the percentages of cement, accelerator and of water, and the testing time. In order to construct a neural network model, a set of 1000 data was used, which were previously obtained based on conventional statistical regression equations derived earlier by the authors. The developed neural networks demonstrated a high performance in the prediction of mold characteristics, in particular a genetic algorithm-trained neural network [24]. Surekha et al. dealt with a similar problem of the mold system using a genetic fuzzy system. A binary-coded genetic algorithm (GA) was used for the optimization of the knowledge base and the rule base. This approach has shown good accuracy in prediction [25].

The proper and complete filling of the mold is the desired result of the sand casting process. The role of casting system design—which consists of a gating system, a feeding system and process parameters—is very significant in achieving this goal. The improper design of the casting system leads to numerous defects in the casting process, resulting in the low quality of the casting part. Casting system design is a complex task that requires expert knowledge of the process. This applied knowledge is often the result of numerous implemented optimizations, modeling, and simulation processes. Artificial intelligence techniques have made a significant contribution to the improvement of the design of casting systems.

Kotas et al. presented a multi-objective optimization problem in the gravity sand casting process. In their study, the authors emphasized a combined approach of software for the casting process simulation and a genetic algorithm (GA) as optimization techniques. The optimization goal was to minimize the top feeder volume, minimize the shrinkage porosity, and limit the centerline porosity, utilizing an optimized arrangement of the chills [26]. Chen et al. conducted research in which they elaborated on an efficient method to reduce the casting defects and upgrade the casting quality. As they intended to minimize the parameters such as the solidification time, filling time, and oxide ratio, the optimized values used were the pouring temperatures, riser diameter, pouring speed, riser position, and pouring diameter. The QPSO algorithm (an improved PSO algorithm) was utilized as an optimization technique, and it provided the reduction of the filling time, solidification time and oxide ratio by 68.14%, 50.56% and 20.20%, respectively [27]. Dučić et al. presented a genetic optimization of the feeder geometry. The optimization process setting (the creation of the fitness function and the appropriate constraints) were based on respecting some rules for the feeding casting part proposed by Campbell [28,29]. With the main objective to minimize the casting process time, Dučić et al. developed a methodology of optimization for the gating system for sand casting using the genetic algorithm (GA). The optimization focus was on the geometry of the gating system, i.e., the cross-section of the ingate and the casting height. In order to verify the validity of the optimized geometry of the gating system, software MAGMA5 was utilized for a numerical simulation [30]. Furthermore, Dučić et al. presented a laboratory setting that promoted fuzzy and neuro-fuzzy intelligent systems for the automatic control of mold filling employed in casting plants. The concept of precision mold filling presupposed three key points in the process, i.e., the precise pouring of the stream into the basin, the maintenance of a constant level of molten metal in the basin, and finally the elimination of the overflow of molten metal from the mold [31]. Ktari and Elmansori optimized the gating system design for smart 3D sand casting by using a relatively new methodology bridging FEM and an Artificial Neural Network (ANN). Their conclusion was that the developed neural network could quickly and successfully predict the ingate velocity for any combination of the significant gating system design parameters [32]. Table 1 presents a brief overview of the analyzed studies of sand casting process improvement.

**Table 1.** Artificial intelligence methods used for the improvement of the sand casting process.

| Authors, Year | Short Description | Artificial Intelligence Methods | | | | |
|---|---|---|---|---|---|---|
| | | GA | PSO | ANN | FL | ANFIS |
| Karunakar and Datta (2007) | Optimization of the composition of green sand mixture. | ● | | ● | | |
| Karunakar and Datta (2007) | Prediction of the major casting part defects. | | | ● | | |
| Kotas et al. (2010) | Optimization of the gravity sand casting process. | ● | | | | |
| Surekha et al. (2011) | Optimization of the green sand mold system. | ● | | | ● | |
| Surekha et al. (2013) | Optimization of the green sand mold system. | ● | ● | | | |
| Parappagoudar et al. (2013) | Prediction of the sand mold characteristics. | ● | | ● | | |
| Chen et al. (2016) | Reduction of the defect of casting part and improve the quality of casting part. | | ● | | | |
| Dučić et al. (2016) | Optimization of the feeder's geometry. | ● | | | | |
| Dučić et al. (2016) | Optimization of the gating system design. | ● | | | | |
| Dučić et al. (2016) | Intelligent systems for automatic control of mold filling. | | | | ● | ● |
| Ktari and Elmansori (2020) | Optimization of the gating system design. | | | ● | | |

### 3.2. Improvement of the Pressure Die-Casting Process

The die-casting process is a quick, reliable, and cost-effective manufacturing process which includes many attributes that contribute to the complexity of the process. The mutual relationships of the process parameters are complex, non-linear, and contradictory. This makes finding their optimal values more complicated. Artificial intelligence techniques have significantly contributed to the improvement of the process through various forms of optimization and modeling.

Rai et al. presented the modeling of the high-pressure die-casting process (HPDC) by using an artificial neural network (ANN). Their ANN model established a correlation between the input values—such as the inlet melt temperature, initial mold temperature, inlet first phase velocity, and inlet second phase velocity—and the output values of the filling time, solidification time and porosity. The development of the ANN model was based on the results obtained from the numerical simulation performed using ProCast software (a FEM-based flow simulation software) [33]. Tsoukalas presented a model based on the genetic algorithm's (GA) application in the definition of the optimal process conditions in pressure die casting. His study aimed to minimize the porosity in the casting part—aluminum alloy $AlSi_9Cu_3$ [34]. Zheng et al. introduced an evaluation system to quantify the surface defects of the casting part. Their solution referred to the high-pressure die-casting process (HPDC). The artificial neural network was used to establish the correlation between surface defects and die casting parameters, such as the mold temperature, pouring temperature, and injection velocity. The developed neural network showed great ability in prediction, and was used to find the optimal process parameters. The optimal parameters were employed, and casting parts with acceptable surface quality were achieved [35]. For the selection of the optimal conditions in the high-pressure die-casting (HPDC) process, Tsoukalas designed an efficient and reliable methodological tool. An adaptive neuro-fuzzy inference system (ANFIS) was applied to investigate the effect of die-casting parameters on porosity formation in AlSi9Cu3 pressure die castings, utilizing the following inputs: the metal temperature, the die temperature, the piston velocity in the low phase, the die gate velocity, and the solidification pressure. Having compared the experimental results with the predicted values, the author suggested that the proposed model should be regarded an efficient tool in defining optimal process conditions in pressure die casting associated with the minimum porosity percentage [36]. Zhang and Wang combined artificial neural networks and genetic algorithms (ANN/GA) to optimize the low-pressure die-cast (LPDC) process. In order to establish the correlation between the process parameters and the casting part quality, an artificial neural network was utilized. The data which were used to build the neural network models were obtained by the numerical simulation of the process. In order to optimize the process parameters with the fitness function based on the trained ANN model, the genetic algorithm was employed [37]. Kittur et al. used a back-propagation neural network (BPNN) algorithm for the modeling of the high pressure die-casting (HPDC) process. They established a correlation between the process parameters—the fast shot velocity, the intensification pressure, the phase change-over point and the holding time—and the casting part characteristics of the surface roughness, hardness and porosity. The results showed that the BPNN approach was able to carry out both the forward and reverse mappings effectively, and can be used in the foundries [38]. Patel et al. represented the forward and reverse modeling of the squeeze casting process based on neural networks (ANN). The developed ANN models successfully established a correlation between process parameters—such as the pressure duration, squeeze pressure, and pouring and die temperatures—and important quality characteristics of the casting part, such as the surface roughness and tensile strength. They also used three population-based search and optimization methods: the genetic algorithm (GA), particle swarm optimization (PSO), and multi-objective particle swarm optimization based on crowding distance (MOPSO-CD) to optimize multiple outputs simultaneously [39,40]. The same group of authors—Patel et al.—conducted a similar study by following other output sizes, such as the density, hardness and secondary dendrite arm spacing. They used different

types of neural networks such as the genetic algorithm neural network (GA-NN), back-propagation neural network (BPNN), and recurrent neural network (RNN) to establish a correlation between the mentioned outputs and inputs, which were represented by process parameters such as the squeeze pressure, pressure duration, and die and pouring temperatures. All of the developed models were able to make effective predictions. The obtained results will help the foundry personnel to automate and precisely control the squeeze casting process [41]. Natrayan and Kumar optimized the squeeze casting process using an artificial neural network. Their study investigated the influence of the squeeze casting process parameters on $AA6061/Al_2O_3/SiC/Gr$ hybrid metal matrix composite. The inputs to the neural network were the process parameters—the squeeze pressure, melt temperature, die temperature and pressure holding time—and the outputs were the mechanical characteristics: the hardness and tensile strength. The neural network has shown great potential in the prediction of mechanical properties (hardness and tensile strength). The neural network showed great potential in the prediction of hardness and tensile strength, such that its success rate was 95% [42]. Lee et al. presented a smart-factory platform for the die casting industry. The mentioned platform is based on three core points: (1) cost-effective product-tracking technology, (2) an advanced process data acquisition system, and (3) a fault detection module based on an artificial neural network. The developed fault detection module based on an artificial neural network showed 96.9 % accuracy for untrained data [43]. Table 2 presents a brief overview of the analyzed studies of pressure die-casting process improvement.

**Table 2.** Artificial intelligence methods used for the improvement of the die-casting process.

| Authors, Year | Short Description | Artificial Intelligence Methods | | | | |
|---|---|---|---|---|---|---|
| | | GA | PSO | ANN | FL | ANFIS |
| Rai et al. (2008) | Modeling of the high pressure die-casting (HPDC) process. | | | ● | | |
| Tsoukalas (2008) | Optimization of the conditions in pressure die-casting process. | ● | | | | |
| Zheng et al. (2009) | Evaluation system to quantify surface defects of the casting part. | | | ● | | |
| Tsoukalas (2011) | Optimization of the conditions in the high-pressure die-casting (HPDC) process. | | | | | ● |
| Zhang and Wang (2012) | Optimization of the low-pressure die-cast (LPDC) process. | ● | | ● | | |
| Patel et al. (2016) | Modeling of the squeeze casting process. | | | ● | | |
| Patel et al. (2016) | Optimization of the squeeze casting process. | ● | ● | | | |
| Kittur et al. (2016) | Modeling the high pressure die-casting (HPDC) process | | | ● | | |
| Patel et al. (2017) | Modelling and analysis of the squeeze casting process. | ● | | ● | | |
| Natrayan and Kumar (2020) | Optimization of the squeeze casting process | | | ● | | |
| Lee et al. (2021) | Smart-factory platform for die casting industry | | | ● | | |

### 3.3. Improvement of the Continuous Casting Process

The continuous casting process is a technically complex manufacturing process, although at first sight it does not seem so, and it belongs to the category of highly productive processes. Its great advantage is that it can be fully automated. Exactly this possibility of process automation was a research challenge that included the application of artificial intelligence techniques in order to efficiently control the process.



Bouhouche et al. used an artificial neural network for the successful identification and control of a nonlinear heat transfer model in the continuous casting process. Their research goal was to control the slab surface temperature in a closed loop, using an artificial neural network (ANN) and PID controllers. Their study took into consideration the dynamic surface temperature, which was an important parameter for the final slab quality [44]. Jabri et al. presented research that dealt with mold level control in the continuous casting process. The mold level variations were a serious productivity and quality problem in the continuous casting process. The authors proposed a level control structure based on Astrom's modified Smith predictor. The tuning parameters were designed through a particle swarm optimization (PSO) approach. The simulation results confirmed the efficiency of the presented architecture [45]. Chen and Huang developed self-organizing fuzzy control (SOFC) to control the molten steel level of a strip casting process. The strip casting process is a steel-strip manufacturing method that combines continuous casting and hot-rolling processes in a single operation. The process is characterized by complex relationships between the process parameters (such as the molten steel level in the tundish, the solidification position and the roll gap), such that it is difficult to establish a control system of a process. The authors presented a solution to this problem as an intelligent controller that can be easily implemented [46]. Jiang et al. established a correlation between the continuous casting technological parameters and the cooling rate of the slab for spring steel by using a BP neural network model. On this basis, the relevant secondary dendrite arm spacing was calculated. The proposed neural network model showed significantly smaller absolute and relative errors compared to the traditional methods [47]. Zhang et al. investigated the effectiveness of a genetic algorithm-based backpropagation (GABP) neural network model and its application to the breakout prediction in the continuous casting process [48]. Tirian et al. presented an adaptive control system for continuous steel casting based on artificial neural networks (ANNs) and fuzzy logic (FL). The neural system (consisting of two distinct neural networks) for the estimation of the crack detection probability was designed, implemented, tested, and integrated into an adaptive control system, which was based on fuzzy logic [49]. Chen et al., in their study, found a fuzzy method for the estimation of the heat flux distribution at the metal–mold interface in slab continuous casting. The deviations between the calculated and measured temperatures acquired with the thermocouples buried in the mold were taken as the input parameters of the fuzzy system [50]. Wang et al. showcased a combined approach for the optimization of the cooling strategy and solidification process in continuous casting. The mentioned approach was based on a common application of the particle swarm optimization (PSO) algorithm, a mathematical heat transfer model and the experimental temperature to determine the heat transfer coefficient. The precise assessment of the heat transfer coefficient of the secondary cooling zone is of the utmost significance because secondary cooling control is the key factor for stabilizing and enhancing the slab quality in continuous casting [51]. Feng et al. presented the application of fuzzy logic and a PSO algorithm in the design of fuzzy PID controller for the control of the mold level in the continuous casting production process. PSO algorithms are used for the optimization of fuzzy controller parameters. The developed methodology was confirmed by numerous simulation studies, and was finally implemented in real industrial conditions [52]. Xu et al., in their study, considered that due to the high-temperature characteristics of SCC (steelmaking-continuous casting) production, the temperature drop deriving from the non-processing process could directly affect energy losses, and could lead to the increase of the energy consumption of each procedure, which greatly influences the overall energy consumption. Following the analyzed problem, they presented the application of the multi-objective hybrid genetic algorithm to minimize the penalty of due date deviation and the extra energy consumption measured by the temperature drop [53]. Table 3 presents a brief overview of the analyzed studies of continuous casting process improvement.

**Table 3.** Artificial intelligence methods used for the improvement of the continuous casting process.

| Authors, Year | Short Description | Artificial Intelligence Methods | | | | |
|---|---|---|---|---|---|---|
| | | GA | PSO | ANN | FL | ANFIS |
| Bouhouche et al. (2008) | Identification and the control of a nonlinear heat transfer model. | | | • | | |
| Jabri et al. (2009) | Mold level control. | | • | | | |
| Chen and Huang (2011) | Control of the molten steel level of a strip casting process. | | | | • | |
| Jiang et al. (2011) | Modeling of the correlation between continuous casting technological parameters and cooling rate. | | | • | | |
| Zhang et al. (2012) | Breakout prediction in the continuous casting process. | • | | • | | |
| Tirian et al. (2014) | Control system for continuous steel casting process. | | | • | • | |
| Chen et al. (2014) | Estimation of the heat flux distribution at the metal-mold interface in slab continuous casting process. | | | | • | |
| Wang et al. (2016) | Optimization of cooling strategy and solidification process in continuous casting process. | | • | | | |
| Feng et al. (2020) | Control of the mold level in the continuous casting process. | | • | | • | |
| Xu et al. (2020) | Optimization of the steelmaking-continuous casting process. | • | | | | |

### 3.4. Improvement of the Investment Casting Process

Investment casting is a very popular and applicable casting technique in present-day industry. Its development is in direct correlation with the aircraft industry, as the production of individual components for the aircraft industry gave investment casting the foremost status in the casting industry. This casting technology is also characterized by nonlinear and complex relationships between the process parameters and characteristics that reflect the quality of the casting part, such that the application of artificial intelligence techniques has a great benefit.

Vosniakos et al. presented a concept that includes a series of neural networks (ANNs) in order to precisely plan the investment casting process. Their starting point was that numerical simulations were not helpful for engineers for workpieces other than the ones simulated. In accordance with that was their attempt to perform a generalization using artificial neural networks (ANN), by taking into account a large number of process parameters (the mold temperature, melting temperature, casting part material, number and location of feeding points, diameter and length of inflow channels, and the angle of channel with respect to the main sprue axis, etc.). The development of the artificial neural networks architectures was based on many precisely defined simulations [54]. Pattnaik et al. presented the optimization of the injection process parameters with multiple performance features in the investment casting process using the orthogonal array with grey–fuzzy logic. The parameters that were subject to optimization were the injection temperature, injection time, and injection pressure, and the accompanied performances were the linear shrinkage and surface finish. The authors presented grey-fuzzy logic as an effective tool to obtain an optimal combination of the process parameters, and their general conclusion was that the quality of wax patterns in the investment casting process can be significantly improved through this approach [55]. In order to establish the correlation between the mechanical features of investment castings on the one side and process parameters and chemical composition on the other side, Sata and Ravi used ANN. The data of related process parameters (shell making, dewaxing, wax making, melting, etc.), the chemical

composition of the alloy, and the resulting mechanical properties (yield strength, ultimate tensile strength, and percentage elongation) for 800 heats were collected in an industrial investment casting foundry. The study was based on the analysis of three different models of ANN (backpropagation, momentum and adaptive, and Levenberg–Marquardt). The authors' findings imply that ANN is adequate for the prediction of mechanical properties [56]. Pattnaik and Kumar used a genetic algorithm (GA) to improve the quality characteristic (surface finish) of the wax patterns in the investment casting process, with the optimized parameters in the surface finish function such as the holding time, injection temperature and die temperature [57]. Sata presented the successful use of ANN to predict a wider range of investment casting defects based on real-time industrial data. The data of 24 parameters related to the process, chemical composition, and defects were collected from about 500 heats in an industrial investment casting foundry. The best results in the establishment of the correlation between the process parameters and investment casting defects were shown by ANN with a Levenberg and Marquardt algorithm [58]. Table 4 presents a brief overview of the analyzed studies of investment casting process improvement.

**Table 4.** Artificial intelligence methods used to improve the investment casting process.

| Authors, Year | Short Description | Artificial Intelligence Methods | | | | |
|---|---|---|---|---|---|---|
| | | **GA** | **PSO** | **ANN** | **FL** | **ANFIS** |
| Vosniakos et al. (2009) | Intelligent precisely planning of the investment casting process. | | | ● | | |
| Pattnaik et al. (2012) | Optimization of the process parameters in the investment casting process. | | | | | ● |
| Sata and Ravi (2014) | Modeling of the investment casting process. | | | ● | | |
| Pattnaik and Kumar (2015) | Optimization of the process parameters in the investment casting process. | ● | | | | |
| Sata (2016) | Modeling of the investment casting process. | | | ● | | |

## 4. Conclusions

This paper provides a detailed overview of the application of five artificial intelligence techniques (artificial neural network (ANN), genetic algorithm (GA), particle swarm optimization (PSO), fuzzy logic (FL) and the adaptive network-based fuzzy inference system (ANFIS)) at the improvement of four casting technologies (sand casting, pressure die casting, continuous casting and investment casting). The application of artificial intelligence techniques has a diverse nature, depending on the specificity of the casting technology. In the improvement of the sand casting process, most studies deal with the modeling and optimization of the design of the green sand mold system, the geometry of the gating system, the feeders, and the process parameters. The most dominant applied artificial intelligence technique is the genetic algorithm (GA), which was used in 63% of the analyzed studies. The artificial intelligence techniques applied to improve the pressure die casting process, mainly through modeling and optimization, established the relationship between the process parameters and the characteristics of the casting part, in order to obtain a high-quality casting part. The most frequent was the application of artificial neural networks, with 72% participation in the analyzed studies. In the continuous casting process in the analyzed studies, artificial intelligence techniques were mainly used in the improvement control process. The most widespread artificial intelligence techniques in these studies were artificial neural networks and fuzzy logic, with 40% participation each in the analyzed studies. The studies related to the improvement of the investment casting process presented the application of artificial intelligence techniques through modeling and optimization, in order to establish a correlation between the process parameters and the characteristics which reflect the quality of the casting part. The most dominant applied

artificial intelligence technique was the artificial neural network (ANN), which was used in 60% of the analyzed studies.

In general, a representation of each artificial intelligence technique separately in all analyzed studies is as follows: ANN—51.3%, GA—37.8%, FL—18.9%, PSO—16.2%, and ANFIS—5.4%. All of the analyzed studies gave a clear overview of the possibility of the application of artificial intelligence techniques in the improvement of the casting process. In addition to the aforementioned clear review of possibilities, the analyzed studies provide a basis for new ideas for researchers and engineers. The results of the conducted research have shown the great potential for the application of artificial intelligence as an integral part of the concept called Industry 4.0 in the development of digitized smart foundries.

**Author Contributions:** Conceptualization: N.D. and S.M.; Methodology: N.D. and Ž.Ć.; Resources: R.R. and A.J.; Formal Analysis: R.R. and A.J.; Validation: S.M. and Ž.Ć.; Writing—Original Draft Preparation: A.J.; Writing—Review and Editing: N.D. All authors have read and agreed to the published version of the manuscript.

**Funding:** This paper is the result of research funded by the Ministry of Education, Science and Technological Development of the Republic of Serbia.

**Institutional Review Board Statement:** Not applicable.

**Informed Consent Statement:** Not applicable.

**Data Availability Statement:** Data sharing is not applicable to this article.

**Conflicts of Interest:** The authors declare no conflict of interest.

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
