# Peer review of "Casting Process Improvement by the Application of Artificial Intelligence"

_applsci, doi:10.3390/app12073264_

Round 1

Reviewer 1 Report

Review of applsci-1594121-peer-review-v1: “Casting process improvement by application of artificial intelligence”

The subject of the paper is relevant with the topics of the journal. As a review paper in the area of the casting process improvements when AI methods are implemented, can attract a great deal of attention from the researchers in the area.

The paper is well structured and separated in paragraphs that gradually achieve the two goals set at the beginning.

It would increase the quality of the paper if the authors were willing to incorporate the following:

  • In table 1, 2, 3 and 4: a separate column describing the optimization characteristics achieved would offer increased readability, in order to have an overview of the questions dealt with in these papers.

My proposal to the editors is to ask the authors for a minor revision of their paper.

Author Response

Dear reviewer,

thank you for the comments you gave us.
We agree with your suggestion that tables (1-4) be supplemented with additional descriptions. That is very, very useful advice, thank you for that.

Adding the results achieved by applying artificial intelligence to each study would make the tables robust. We have therefore added a column in which only one sentence is given, which says what the study covers. We hope that this has increased the effectiveness of the tables.

Changes are highlighted in green in the text.

Best regards,
Authors 

Reviewer 2 Report

The paper presents an overview of conducted research studies, which deal with the appli-15 cation of artificial intelligence at improving the casting process. This manuscript is goog enough to push the recognition of artificial intelligence in the industry field. The cited references are up-to-date, and the language organization is well. So, I recommend this manuscript.

Author Response

Dear reviewer,

thank you for the comments you gave us.

Best regards,
Authors 

Reviewer 3 Report

The introduction of artificial intelligence into industrial spheres significantly increases the foundry's ability not only on a domestic scale, but also on a global scale.

For this reason, I consider such an overview of the work done so far to be a relevant contribution in the field.

Please authors review the article and edit any typos.

Example: cast and casting

  • casting - is a technological process for the production of components
  • cast - cast - is a product made by casting

Author Response

Dear reviewer,

thank you for the comments you gave us.

We have made corrections in the paper you suggested. 

Technology is: casting or casting process.
Product is casting part.

We think that now there will be no confusion in understanding the text on the relation technology (casting or casting process) - product (casting part).

Changes are highlighted in green.

Best regards,
Authors